# On Provably Robust Meta-Bayesian Optimization

**Zhongxiang Dai**[1]    **Yizhou Chen**[1]    **Haibin Yu**[2]    **Bryan Kian Hsiang Low**[1]    **Patrick Jaillet**[3]

[1]Department of Computer Science, National University of Singapore, Republic of Singapore
[2]Department of Data Platform, Tencent
[3]Department of Electrical Engineering and Computer Science, Massachusetts Institute of Technology, USA

## Abstract

*Bayesian optimization* (BO) has become popular for sequential optimization of black-box functions. When BO is used to optimize a target function, we often have access to previous evaluations of potentially related functions. This begs the question as to whether we can leverage these previous experiences to accelerate the current BO task through *meta-learning* (meta-BO), while ensuring *robustness* against potentially harmful dissimilar tasks that could sabotage the convergence of BO. This paper introduces two scalable and provably robust meta-BO algorithms: *robust meta-Gaussian process-upper confidence bound* (RM-GP-UCB) and *RM-GP-Thompson sampling* (RM-GP-TS). We prove that both algorithms are asymptotically no-regret even when some or all previous tasks are dissimilar to the current task, and show that RM-GP-UCB enjoys a better theoretical robustness than RM-GP-TS. We also exploit the theoretical guarantees to optimize the weights assigned to individual previous tasks through regret minimization via online learning, which diminishes the impact of dissimilar tasks and hence further enhances the robustness. Empirical evaluations show that (a) RM-GP-UCB performs effectively and consistently across various applications, and (b) RM-GP-TS, despite being less robust than RM-GP-UCB both in theory and in practice, performs competitively in some scenarios with less dissimilar tasks and is more computationally efficient.

## 1 INTRODUCTION

*Bayesian optimization* (BO) has recently gained immense popularity as an efficient method to optimize black-box functions [Shahriari et al., 2016], and it has found success in a variety of applications such as automated *machine learning* (ML) [Snoek et al., 2012], *reinforcement learning* (RL) [Wilson et al., 2014], among others. BO uses a *Gaussian process* (GP) [Rasmussen and Williams, 2006] as a surrogate to represent the belief about the objective function and, in each iteration, queries the input parameters that maximize an *acquisition function*. In particular, the BO algorithms based on the *GP-upper confidence bound* (GP-UCB) [Srinivas et al., 2010] and *GP-Thompson sampling* (GP-TS) [Chowdhury and Gopalan, 2017] acquisition functions have been shown to be asymptotically *no-regret* and perform competitively in practice. When using BO to optimize a *target function*, we sometimes have access to a set of evaluations of some potentially related functions. For example, when using BO for hyperparameter optimization of an ML model trained on a target dataset, we often have access to some previously completed BO tasks using other potentially related datasets [Golovin et al., 2017]. These previous tasks, if similar to the current task, may be exploited to accelerate the current BO task. However, if some (or even all) previous tasks are in fact dissimilar to the current task, their use may turn out to incorporate harmful information and sabotage the convergence of BO [Feurer et al., 2018]. This begs the question as to whether we can leverage previous tasks to improve the efficiency of the current BO task, while ensuring *robustness* against harmful dissimilar tasks such that they do not affect the trademark *no-regret* convergence of BO.

Exploiting previous learning experiences to improve the efficiency of the current task is the goal of *meta-learning* [Vanschoren, 2018]. Meta-learning is a broad field with various applications in supervised learning [Finn et al., 2017], RL [Xu et al., 2018], active learning [Pang et al., 2018], among others. The major challenges in meta-learning include (a) the transfer of information from previous tasks to the current task, and (b) characterization of task similarity which is crucial for identifying harmful dissimilar tasks [Vanschoren, 2018]. The application of meta-learning to BO (or *meta-BO*) has been explored by previous studies which differ in how these two challenges are addressed.

*Accepted for the 38th Conference on Uncertainty in Artificial Intelligence* (UAI 2022).

Some works, such as multitask BO [Swersky et al., 2013], transfer the information from previous tasks by building a joint GP surrogate using the observations from all previous and current tasks, with the task similarity either represented by meta-features [Bardenet et al., 2013, Yogatama and Mann, 2014] or learned from observations [Swersky et al., 2013, Wang et al., 2018]. These works, however, are limited by the scalability of GP due to including all previous and current observations in a single GP [Feurer et al., 2018].[1] To this end, other recent works transfer information from previous tasks using a more scalable approach: They build a separate GP surrogate for each individual task and use a weighted combination of either the individual surrogate functions or acquisition functions for query selection [Feurer et al., 2018, Wistuba et al., 2016, 2018]. A more detailed review of related works is presented in Sec. 7. However, none of the previous works has provided a theoretical performance guarantee to ensure robust performances in the presence of harmful dissimilar tasks. A robust theoretical guarantee is important for guaranteeing the consistent performances of meta-BO algorithms in various real-world applications, which is crucial for their practical deployment.

To this end, this paper introduces two scalable and provably robust meta-BO algorithms: *robust meta-GP-upper confidence bound* (RM-GP-UCB) and *robust meta-GP-Thompson sampling* (RM-GP-TS). Both algorithms compute the acquisition function (GP-UCB or GP-TS) for each individual task and select the next query via either a weighted combination (RM-GP-UCB) or in a probabilistic way (RM-GP-TS) (Sec. 3). As a result, like the works of Feurer et al. [2018], Wistuba et al. [2016, 2018], a separate GP surrogate is built for each previous task, making our algorithms scale well in the number of meta-tasks and observations in each meta-task. Our major contributions include: **Firstly**, we prove robust theoretical convergence guarantees for both RM-GP-UCB and RM-GP-TS (Sec. 4). In particular, both algorithms are asymptotically *no-regret* for *any* given set of previous tasks, i.e., even if some or all previous tasks are dissimilar to the target task. Moreover, we show that RM-GP-UCB enjoys a superior robustness guarantee compared with RM-GP-TS (Sec. 4.2). **Secondly**, to further enhance our robustness against dissimilar tasks, we exploit the theoretical guarantees to learn the task similarity (and hence identify dissimilar tasks) in a principled way, by minimizing the regret upper bounds via a computationally cheap online learning algorithm known as *Follow-The-Regularized-Leader* (Sec. 5). **Lastly**, we use extensive empirical evaluations to show that: RM-GP-UCB performs effectively and consistently across a wide range of tasks; RM-GP-TS, despite under-performing in adverse scenarios (i.e., when a large number of previous tasks are dissimilar),

performs competitively in some favorable cases with less dissimilar tasks and is much more computationally efficient. Of note, our theoretical and empirical comparisons between RM-GP-UCB and RM-GP-TS may provide useful insights for other meta-BO algorithms in general (and potentially for other related algorithms such as meta-RL) in terms of the relative strengths and weaknesses of UCB- and TS-based meta-learning algorithms.

# 2 BACKGROUND AND PROBLEM FORMULATION

**Bayesian Optimization.** This work tackles the problem of sequentially maximizing an unknown function $f : \mathcal{D} \to \mathbb{R}$. In each iteration $t = 1, \ldots, T$, an input $\mathbf{x}_t \in \mathcal{D}$ (a $D \geq 1$-dimensional vector) is queried to yield $y_t \triangleq f(\mathbf{x}_t) + \epsilon$ where $\epsilon \sim \mathcal{N}(0, \sigma^2)$ is a Gaussian noise with variance $\sigma^2$. The performance of BO is typically measured by *cumulative regret*: $R_T \triangleq \sum_{t=1,\ldots,T} [f(\mathbf{x}^*) - f(\mathbf{x}_t)]$ where $\mathbf{x}^* \in \arg\max_{\mathbf{x} \in \mathcal{D}} f(\mathbf{x})$ is a global maximizer of $f$. It is desirable for a BO algorithm to achieve *no regret* by making its $R_T$ grow sublinearly such that its *simple regret* $S_T \triangleq \min_{t=1,\ldots,T} [f(\mathbf{x}^*) - f(\mathbf{x}_t)] \leq R_T/T$ goes to 0 asymptotically. During BO, we model the belief about $f$ using a *Gaussian process* (GP) $\{f(\mathbf{x})\}_{\mathbf{x} \in \mathcal{D}}$. That is, any finite subset of $\{f(\mathbf{x})\}_{\mathbf{x} \in \mathcal{D}}$ follows a multivariate Gaussian distribution [Rasmussen and Williams, 2006]. A GP is fully specified by its prior mean $\mu(\mathbf{x})$ and kernel function $k(\mathbf{x}, \mathbf{x}')$, and we assume w.l.o.g. that $\mu(\mathbf{x}) = 0$ and $k(\mathbf{x}, \mathbf{x}') \leq 1 \; \forall \mathbf{x}, \mathbf{x}' \in \mathcal{D}$. We focus on the widely used Squared Exponential (SE) kernel. Given $T$ noisy observations $\mathbf{y}_T \triangleq [y_t]_{t=1,\ldots,T}^\top$ at inputs $\mathbf{x}_1, \ldots, \mathbf{x}_T$, the posterior GP belief of $f$ at input $\mathbf{x} \in \mathcal{D}$ is Gaussian with the following posterior mean and variance:

$$\mu_T(\mathbf{x}) \triangleq \mathbf{k}_T(\mathbf{x})^\top (\mathbf{K}_T + \lambda I)^{-1} \mathbf{y}_T,$$
$$\sigma_T^2(\mathbf{x}) \triangleq k(\mathbf{x}, \mathbf{x}) - \mathbf{k}_T(\mathbf{x})^\top (\mathbf{K}_T + \lambda I)^{-1} \mathbf{k}_T(\mathbf{x}), \quad (1)$$

where $\mathbf{K}_T \triangleq [k(\mathbf{x}_t, \mathbf{x}_{t'})]_{t,t'=1,\ldots,T}$, $\mathbf{k}_T(\mathbf{x}) \triangleq [k(\mathbf{x}_t, \mathbf{x})]_{t=1,\ldots,T}^\top$, $\lambda$ is a regularization parameter.

**Meta-Bayesian Optimization.** We refer to the function $f$ being maximized as the *target function* and the functions $f_i$ for $i = 1, \ldots, M$ of the $M$ previous tasks as *meta-functions*. We use *target task/observations* and *meta-tasks/observations* in a similar manner. All functions are defined on the same domain $\mathcal{D}$ which is assumed to be discrete for simplicity, but the theoretical results can be easily generalized to continuous domains following the analysis of previous works [Chowdhury and Gopalan, 2017, Srinivas et al., 2010]. We assume that $f$ and all $f_i$'s lie in the *reproducing kernel Hilbert space* (RKHS) associated with the kernel $k$ such that their norm induced by the RKHS is bounded: $\|f\|_k \leq B, \|f_i\|_k \leq B, \forall i = 1, \ldots, M$. This assumption intuitively suggests that the target and meta-functions have

---

[1]Some works such as Perrone et al. [2018] and Volpp et al. [2020] replace GP by other surrogate models such as neural networks for scalability, however, they lack the principled uncertainty estimate and theoretical guarantee offered by GP.

the same degree of smoothness. Same as the work of Wang et al. [2018] which has also performed theoretical analysis of a meta-learning algorithm for BO, we also assume that all meta- and target observations are corrupted by a Gaussian noise $\epsilon \sim \mathcal{N}(0, \sigma^2)$ with variance $\sigma^2$. The number of observations from meta-task $i$ is a constant denoted as $N_i$, and $N \triangleq \max_{i=1,\ldots,M} N_i$. $\mathbf{x}_{i,j}$ and $y_{i,j}$ represent the $j$-th input and noisy output of meta-task $i$ respectively. We define the *function gap* $d_i \triangleq \max_{j=1,\ldots,N_i} \left| f(\mathbf{x}_{i,j}) - f_i(\mathbf{x}_{i,j}) \right| < \infty$ which represents the maximum difference between the function values of $f$ and $f_i$ at any corresponding input $\mathbf{x}_{i,j}$ of meta-task $i$. Note that for a given set of meta-observations for meta-task $i$, the function gap $d_i$ is an unknown constant characterizing the similarity between meta-task $i$ and the target task: a smaller function gap implies a stronger similarity.

## 3 ROBUST META-BAYESIAN OPTIMIZATION

The acquisition function (2) adopted by RM-GP-UCB in iteration $t$ is a weighted combination of $M + 1$ individual GP-UCB acquisition functions [Srinivas et al., 2010] for the target task and the $M$ meta-tasks, each of which is calculated using the observations from a particular task:

$$
\overline{\zeta}_t^{\text{UCB}}(\mathbf{x}) \triangleq \nu_t \left[ \sum_{i=1}^{M} \omega_i \left[ \overline{\mu}_i(\mathbf{x}) + \tau \overline{\sigma}_i(\mathbf{x}) \right] \right] + \\ (1 - \nu_t) \left[ \mu_{t-1}(\mathbf{x}) + \beta_t \sigma_{t-1}(\mathbf{x}) \right] . \quad (2)
$$

In (2), $\mu_{t-1}(\mathbf{x})$ and $\sigma_{t-1}(\mathbf{x})$ represent, respectively, the GP posterior mean and standard deviation (1) at $\mathbf{x}$ calculated using the target observations from iterations 1 to $t-1$. $\overline{\mu}_i(\mathbf{x})$ and $\overline{\sigma}_i(\mathbf{x})$ are computed using all meta-observations from meta-task $i$. $\beta_t > 0$ and $\tau > 0$ will be defined in Sec. 4. $\nu_t \in [0, 1]$ can be interpreted as the overall weight given to all meta-tasks in iteration $t$ and should be chosen to be non-increasing in $t$, which enforces the impact of meta-tasks in (2) to be non-increasing. The *meta-weights* $\omega_i$'s can be understood as the weights assigned to individual meta-tasks. Note that since the dataset used to calculate $\overline{\mu}_i(\mathbf{x})$ and $\overline{\sigma}_i(\mathbf{x})$ is fixed with size $N_i$, the matrix inversion in (1) (i.e., the computational bottleneck for GP) can be pre-computed. So, after $T$ iterations, RM-GP-UCB incurs $\mathcal{O}(T^3)$ time for covariance matrix inversion (since only the target covariance matrix of size $T \times T$ needs to be inverted) and $\mathcal{O}(MN^2 + T^2)$ time during predictive inference, which are less than the respective $\mathcal{O}((MN+T)^3)$ and $\mathcal{O}((MN+T)^2)$ time when all observations are included in a single GP. In practice, the total number of BO iterations ($T$) is usually small, therefore, the differences between these corresponding computational costs can be large, especially when $M$ and $N$ are large. Hence, RM-GP-UCB is scalable in the number of meta-tasks ($M$) and observations in each meta-task ($N$).

The acquisition function of RM-GP-TS is defined as:

$$
\overline{\zeta}_t^{\text{TS}}(\mathbf{x}) \triangleq \begin{cases} f^t(\mathbf{x}) & \text{with probability } 1 - \nu_t , \\ \sum_{i=1}^{M} \omega_i \overline{f}_i^t(\mathbf{x}) & \text{with probability } \nu_t, \end{cases} \quad (3)
$$

in which $f^t$ is a function sampled from the GP posterior of the target task: $f^t \sim \mathcal{GP}\left(\mu_{t-1}(\cdot), \beta_t^2 \sigma_{t-1}^2(\cdot)\right)$, and $f_i^t$ is sampled from the GP posterior of meta-task $i$: $\overline{f}_i^t \sim \mathcal{GP}\left(\overline{\mu}_i(\cdot), \tau^2 \overline{\sigma}_i^2(\cdot)\right)$. Using approximation techniques such as random Fourier features (RFF) approximation [Rahimi and Recht, 2008] (which we use in all our experiments), the functions $f^t$ and $f_i^t$'s can be sampled efficiently, hence making RM-GP-TS computationally efficient (as we will demonstrate in Sec. 6). Moreover, since the meta-observations of every meta-task is fixed, the use of approximation techniques such as RFF allows the functions $\overline{f}_i^t$'s to be sampled beforehand before the algorithm starts. Refer to Appendix D.5 for more details on RM-GP-TS.

In iteration $t$ of either RM-GP-UCB or RM-GP-TS (Algorithm 1), we first optimize the meta-weights and update $\nu_t$ (Sec. 5.2), which corresponds to line 2 of Algorithm 1. Next, the input $\mathbf{x}_t$ is selected by maximizing the acquisition function (2) (RM-GP-UCB) or (3) (RM-GP-TS), after which we query $\mathbf{x}_t$ and use the newly collected $(\mathbf{x}_t, y_t)$ to update the GP posterior belief (1).

---

**Algorithm 1** RM-GP-UCB/RM-GP-TS
1: **for** $t = 1, 2, \ldots, T$ **do**
2:    Update $\omega_i$ for $i = 1, \ldots, M$ via online meta-weight optimization and update $\nu_t$ (Sec. 5.2)
3:    $\mathbf{x}_t \leftarrow \arg\max_{\mathbf{x} \in \mathcal{D}} \overline{\zeta}_t^{\text{UCB}}(\mathbf{x})$ (for RM-GP-UCB) (2), or $\mathbf{x}_t \leftarrow \arg\max_{\mathbf{x} \in \mathcal{D}} \overline{\zeta}_t^{\text{TS}}(\mathbf{x})$ (for RM-GP-TS) (3)
4:    Query $\mathbf{x}_t$ to observe $y_t$, and update GP posterior belief (1) using $(\mathbf{x}_t, y_t)$
5: **end for**

---

## 4 THEORETICAL ANALYSIS

### 4.1 RM-GP-UCB

Theorem 1 presents an upper bound on the cumulative regret of RM-GP-UCB (proof in Appendix A).

**Theorem 1** (RM-GP-UCB). *Let $\delta \in (0, 1)$. Denote by $\gamma_t$ the maximum information gain about $f$ from observing any set of $t$ observations. If RM-GP-UCB is run with: $\lambda = 1 + 2/T$, $\beta_t = B + \sigma\sqrt{2(\gamma_{t-1} + 1 + \log(4/\delta))}$, $\tau = B + \sigma\sqrt{2(\gamma_N + 1 + \log(4M/\delta))}$, $\nu_t \in [0, 1]$ and $\nu_{t+1} \leq \nu_t$, $\omega_i \geq 0$ and $\sum_{i=1}^{M} \omega_i = 1$. Then, with probability of $\geq 1 - 3\delta/4$,*

$$
R_T \leq 2(\alpha + \tau) \sum_{t=1}^{T} \nu_t + \beta_T \sqrt{C_1 T \gamma_T}
$$

$$= \widetilde{\mathcal{O}}\big(\big(\sum_{i=1}^{M} d_i\big) \sum_{t=1}^{T} \nu_t + \gamma_T \sqrt{T}\big), \qquad (4)$$

*where* $C_1 \triangleq \frac{8}{1+\sigma^{-2}}$, *and* $\alpha \triangleq \sum_{i=1}^{M} \omega_i \frac{N_i}{\sigma^2}(2\sqrt{2\sigma^2 \log \frac{8N_i}{\delta}} + d_i)$.

The second term $\gamma_T \sqrt{T}$ in the regret upper bound (4) grows sub-linearly for the SE kernel for which $\gamma_T = \mathcal{O}((\log T)^{D+1})$. Therefore, if $\nu_t$ is designed such that $\nu_t \to 0$ as $t \to \infty$, the first term also grows sub-linearly and hence RM-GP-UCB is asymptotically no-regret.

Theorem 1 holds for a given set of meta-tasks with fixed yet unknown $d_i$'s. Note that we do not impose assumptions on the values of $d_i$'s, i.e., the similarities between the meta- and target tasks. Therefore, Theorem 1 gives a robust regret upper bound which holds for *any* given set of meta-tasks. In other words, even in adverse scenarios where some or all meta-tasks are extremely dissimilar to the target task (i.e., when some or all $d_i$'s are very large), RM-GP-UCB is still asymptotically no-regret, which indicates the robustness and generality of our algorithm. This provides an assurance about the *worst-case behavior* in any given scenario.[2] In our proof, the key step (Lemma 3 in Appendix A) is to upper bound (by $\alpha$ in Theorem 1) the overall error induced by the use of any given set of meta-observations, instead of the target observations at the same corresponding input locations, when calculating the acquisition function (2). These interpretations also explain the dependence of $\alpha$, hence the regret bound, on $d_i$ and $N_i$: Larger function gaps increase the error resulting from the use of the meta-observations, and a larger number of meta-observations also inflates the worst-case upper bound by accumulating the individual errors. Of note, a limitation of our regret upper bound (Theorem 1) is that it does not reflect the benefit of the use of the meta-tasks when they are indeed similar to the target task. Next, we use our theoretical analysis to give some insights on how the meta-tasks, if similar to the target task, help improve the convergence of our algorithm.

**Meta-tasks Can Improve the Convergence by Accelerating Exploration.** In addition to characterizing the worst-case behavior, we also use our theoretical analysis to illustrate how meta-tasks can help RM-GP-UCB converge faster than standard GP-UCB. As we have proved in Appendix A.3, at the early stage of the algorithm, the meta-tasks (if similar to the target task) can help RM-GP-UCB obtain a smaller regret upper bound than GP-UCB by *reducing the uncertainty at the selected input*. Equivalently, the additional information from the meta-tasks allows RM-GP-UCB to *reduce the*

*degree of exploration at the early stage*. Since initial exploration of BO usually incurs large regrets, less exploration results in smaller regrets. At later stages when $\nu_t$ becomes close to 0, RM-GP-UCB converges to no regret at a similar rate to GP-UCB (i.e., the second term $\gamma_T \sqrt{T}$ in the regret upper bound (4) dominates).

## 4.2 RM-GP-TS

Theorem 2 gives an upper bound on the cumulative regret of RM-GP-TS (proof in Appendix B).

**Theorem 2** (RM-GP-TS). *Define* $d'_i \triangleq \max_{\mathbf{x} \in \mathcal{D}} |f(\mathbf{x}) - f_i(\mathbf{x})|$. *With the same parameters as those defined in Theorem 1, we have that with probability of at least* $1 - 3\delta/4$,

$$R_T = \widetilde{\mathcal{O}}\big(\big(\sum_{i=1}^{M} \omega_i d'_i\big) \sum_{t=1}^{T} \nu_t + \sum_{t=1}^{T} \nu_t \sqrt{\gamma_t} + \gamma_T \sqrt{T}\big).$$

Note that by definition, we have that $d'_i \geq d_i, \forall i$. Similar to RM-GP-UCB, as long as $\nu_t$ is chosen such that $\nu_t \to 0$ as $t \to \infty$ and that $\nu_t = o(1/\sqrt{\gamma_t})$, all three terms in Theorem 2 are sub-linear (for the SE kernel). That is, RM-GP-TS is also asymptotically no-regret for any set of meta-tasks, even when some or all meta-tasks are dissimilar to the target task. Moreover, comparing the extra terms in the regret upper bounds resulting from the use of the meta-tasks for both RM-GP-UCB (i.e., the first term of equation (4) in Theorem 1) and RM-GP-TS (i.e., the first two terms of Theorem 2) reveals that compared with RM-GP-UCB, RM-GP-TS suffers from a worse extra dependence on $T$ due to the meta-tasks. Specifically, while the first terms of Theorems 1 and 2 have the same dependence on $T$, the second term of Theorem 2 introduces an extra dependence on $T$ which dominates the first term. This suggests that in adverse scenarios with a large number of dissimilar tasks, RM-GP-TS may suffer from a worse convergence than RM-GP-UCB. In other words, RM-GP-UCB enjoys a better theoretically guaranteed robustness against dissimilar tasks.

## 4.3 PRACTICAL IMPLICATIONS

Besides the theoretical insights, Theorems 1 and 2 also provide two natural hints to the practical algorithmic design. Firstly, note that both Theorems hold for all choices of meta-weights $\omega_i$'s. Therefore, we have the flexibility to choose the optimal $\omega_i$'s (i.e., learn the task similarity) by minimizing the regret upper bounds in Theorems 1 and 2. Secondly, the first term in Theorem 1 suggests that we can lower the regret by making $\nu_t$ (i.e., the influence of the meta-tasks) decay faster if $\alpha$ in Theorem 1 (i.e., an upper bound on the error produced by using the meta-tasks) is larger. The same reasoning applies to Theorem 2, i.e., we can decay $\nu_t$ faster if $\sum_{i=1,...,M} \omega_i d'_i$ in Theorem 2 is larger. Both design choices can further strengthen the robustness of our

---

[2]This notion of robustness is in line with that of *robust optimization* (RO) [Beyer and Sendhoff, 2007] which also attempts to optimize the performance in the worst-case scenario. The difference is that RO optimizes an explicit objective, while we aim at preserving the no-regret property in the worst case.

algorithms against dissimilar meta-tasks by lessening their impact. Unfortunately, they both require the values of the function gaps $d_i$'s which are unavailable.[3] To this end, we devise a principled technique to estimate upper bounds on the function gaps, which is presented in the next section.

# 5 ONLINE META-WEIGHT OPTIMIZATION

In this section, we first introduce a principled technique for estimating high-probability upper bounds on the function gaps (Sec. 5.1) that, when combined with Theorems 1 and 2, naturally yields a principled method for optimizing the meta-weights through regret minimization via online learning.

## 5.1 ONLINE ESTIMATION OF FUNCTION GAPS

Inspired by the confidence region constructed by GP-UCB [Srinivas et al., 2010, Chowdhury and Gopalan, 2017] that contains the target function with high probability, after $t \geq 1$ target observations have been collected, define

$$
\begin{aligned}
U_{t,i,j} &\triangleq \mu_t(\mathbf{x}_{i,j}) + \beta_{t+1}\sigma_t(\mathbf{x}_{i,j}), \\
L_{t,i,j} &\triangleq \mu_t(\mathbf{x}_{i,j}) - \beta_{t+1}\sigma_t(\mathbf{x}_{i,j}),
\end{aligned}
\tag{5}
$$

where $\mathbf{x}_{i,j}$ is the $j$-th input of meta-task $i$, $\beta_{t+1}$ is previously defined in Theorem 1, and $U_{t,i,j}$ and $L_{t,i,j}$ can be interpreted, respectively, as the upper and lower confidence bounds of $f$ at $\mathbf{x}_{i,j}$ after $t$ iterations. Lemma 2 (Appendix A) implies that with probability of at least $1 - \delta/4$ ($\delta$ is defined in Theorem 1): $L_{t,i,j} \leq f(\mathbf{x}_{i,j}) \leq U_{t,i,j}, \forall t, i, j$. Consequently, the following result gives high-probability upper bounds on the function gaps (proof in Appendix C.1):

**Lemma 1.** *With probability of at least $1 - \delta$,*

$$
\begin{aligned}
d_i \leq &\sqrt{2\sigma^2 \log\left[\left(8\sum\nolimits_{i=1}^{M} N_i\right)/\delta\right]} + \\
&\max_{j=1,\ldots,N_i}\left[\max\{|y_{i,j} - U_{t,i,j}|, |y_{i,j} - L_{t,i,j}|\}\right] \triangleq \overline{d}_{i,t},
\end{aligned}
$$

*for $t = 1, \ldots, T$ and $i = 1, \ldots, M$.*

Unlike $d_i$, $\overline{d}_{i,t}$ can be efficiently calculated as its incurred time is linear in both $M$ and $N$.

## 5.2 ONLINE META-WEIGHT OPTIMIZATION THROUGH REGRET MINIMIZATION

In this section, we focus on RM-GP-UCB since the analysis for RM-GP-TS (deferred to Appendix C.4) is similar and leads to the same update rules for $\omega_i$'s and $\nu_t$. Combining Lemma 1 and Theorem 1 allows us to derive the following result for RM-GP-UCB (proof in Appendix C.2):

---

[3] $d_i$ can be used as an estimate of $d_i'$ since $d_i' \geq d_i$ (Sec. 4.2).

**Proposition 1** (RM-GP-UCB). *With probability of $\geq 1 - \delta$,*

$$
\begin{aligned}
R_T \leq &\frac{2}{\sigma^2}\left[\sum\nolimits_{t=1}^{T}\boldsymbol{\omega}^\top \boldsymbol{l}_t\right]\left[\sum\nolimits_{t=1}^{T}\nu_t\right] + \\
&2\tau\sum\nolimits_{t=1}^{T}\nu_t + \beta_T\sqrt{C_1 T \gamma_T},
\end{aligned}
$$

*where $\boldsymbol{\omega} \triangleq [\omega_i]_{i=1,\ldots,M}$, $\boldsymbol{l}_t \triangleq [l_{i,t}]_{i=1,\ldots,M}$, and $l_{i,t} \triangleq N_i(2\sqrt{2\sigma^2 \log(8N_i/\delta)} + \overline{d}_{i,t})$.*

Note that $\boldsymbol{l}_t$ can be efficiently computed after the $t$-th observation is collected. The regret upper bound in Proposition 1 depends on $\omega_i$'s only through the term $\sum_{t=1}^{T}\boldsymbol{\omega}^\top \boldsymbol{l}_t$ which can be minimized to derive the optimal meta-weights. This constitutes an *online learning* problem with linear loss function and its solution $\boldsymbol{\omega}$ constrained to a probability simplex. An additional entropic regularization term is usually preferred so as to encourage a solution with a large entropy to stabilize it [Bubeck, 2011]. This corresponds to encouraging the meta-weights to spread across a large number of meta-tasks, in order to discover as many similar meta-tasks as possible. As a result, by using $1/\eta$ ($\eta > 0$) as the regularization parameter, the optimal $\boldsymbol{\omega}$ in iteration $t > 1$ is obtained by solving the following optimization problem:

$$
\boldsymbol{\omega} \triangleq \arg\min_{\boldsymbol{\omega}'} \sum\nolimits_{s=1}^{t-1}\boldsymbol{\omega}'^\top \boldsymbol{l}_s + \eta^{-1}\sum\nolimits_{i=1}^{M}\omega_i'\log\omega_i', \tag{6}
$$

subject to the constraints: $\omega_i' \geq 0, \forall i$ and $\sum_{i=1}^{M}\omega_i' = 1$. When $t = 1$, the optimal $\boldsymbol{\omega}$ follows from optimizing only the entropic regularization term, thus naturally entailing the uniform distribution $\omega_i = 1/M, \forall i$. Consequently, (6) corresponds exactly to the online learning algorithm called *Follow-The-Regularized-Leader* with an entropic regularizer [Bubeck, 2011] where $\eta$ represents the learning rate. Its optimal solution in iteration $t$ can be derived via Lagrange multiplier (Appendix C.3) as

$$
\omega_i = \frac{e^{-\eta \sum_{s=1}^{t-1} l_{i,s}}}{\sum_{j=1}^{M} e^{-\eta \sum_{s=1}^{t-1} l_{j,s}}} \overset{(a)}{\approx} \frac{e^{-\eta N \sum_{s=1}^{t-1} \overline{d}_{i,s}}}{\sum_{j=1}^{M} e^{-\eta N \sum_{s=1}^{t-1} \overline{d}_{j,s}}}, \tag{7}
$$

for $i = 1, \ldots, M$ where (a) follows from assuming that all $N_i$'s are close to $N$ for simplicity. With this simplification, the first (noise-correction) term in the expression of $\overline{d}_{i,t}$ from Lemma 1 also cancels out, thus leading to a neat and elegant update rule for $\omega_i$ which we use in all our experiments. As is evident from (7), the update of $\omega_i$'s in each iteration only involves computing $\overline{d}_{i,t}$'s (incurring $\mathcal{O}(MN)$ time), adding one term to the summation on the exponent ($\mathcal{O}(M)$ time), and a normalization step ($\mathcal{O}(M)$ time), all of which are computationally cheap. Intuitively, (7) assigns small weights to meta-tasks with a large cumulative estimated function gap which implies a less similar meta-task.

In addition, $\overline{d}_{i,t}$ from Lemma 1 also allows for the estimation of an upper bound on $\alpha$ (Theorem 1) in each iteration (i.e., by simply replacing $d_i$ with $\overline{d}_{i,t}$) and thus facilitates an

adaptive selection of $\nu_t$, as mentioned in Sec. 4. Specifically, we set $\nu_1 = 1$ and $\nu_t = \nu_{t-1} \times \min(r, (\sum_{i=1}^{M} \omega_i \overline{d}_{i,t})^{-\epsilon})$ for $t > 1$, in which we have dropped the constants independent of $\overline{d}_{i,t}$. $r \in (0, 1)$ represents the minimum decaying rate to ensure the monotonic decay of $\nu_t$ such that RM-GP-UCB is no-regret (Sec. 4.1). $\epsilon > 0$ controls the aggressiveness of the adaptive decay such that a larger $\epsilon$ results in a faster decay. With this scheme, when the overall estimated function gaps are larger (the meta-tasks are dissimilar), $\nu_t$ decays faster and thus the impact of the meta-tasks vanishes more quickly.

Importantly, when optimizing the values of $\omega_i$'s and $\nu_t$ as described above, we have taken into account the limitation of our regret upper bounds (i.e., they do not reflect the benefit of the use of the meta-tasks, Sec. 4.1) and hence incorporated additional practical considerations. Specifically, we have optimized the $\omega_i$'s with an additional entropic regularization term to encourage the $\omega_i$'s to spread across a large number of meta-tasks, and optimized $\nu_t$ such that it decreases faster if $\alpha$ (i.e., an upper bound on the error induced by the use of the meta-tasks) is larger.

# 6 EXPERIMENTS AND DISCUSSION

We use extensive real-world experiments to compare our RM-GP-UCB and RM-GP-TS with *(1)* standard GP-UCB, two other GP-based scalable meta-BO algorithms: *(2) ranking-weighted Gaussian process ensemble* (RGPE) [Feurer et al., 2018] and *(3) transfer acquisition function* (TAF) [Wistuba et al., 2018], *(4)* multitask BO (MTBO) [Swersky et al., 2013], and *(5)* the method from [Wang et al., 2018] named *point estimate meta-BO* (PEM-BO). Since MTBO is relatively not scalable (Sec. 1), we only apply it to those experiments with relatively small number of meta-tasks and observations for which MTBO is still computationally feasible. We compare with PEM-BO [Wang et al., 2018] in the experiment that is most favorable for this algorithm, i.e., with the largest number of meta-observations and a discrete domain (refer to Sec. 6.2 for more details). We set $\eta = 1/N$, $\epsilon = 0.7$ and $r = 0.7$ in all real-world experiments to demonstrate the robustness of our algorithm against the choice of these parameters. In practice, the upper bound on the function gap, $\overline{d}_{i,t}$, from Lemma 1 may be too conservative; so, we replace the outer $\max$ operator over $j = 1, ..., N_i$ with the empirical mean in our experiments.[4] Some details and results are deferred to Appendix D due to lack of space. All error bars represent standard errors. Our code is available at `https://github.com/daizhongxiang/meta-BO`.

---

[4]We explore the difference between them in Appendix D.3.

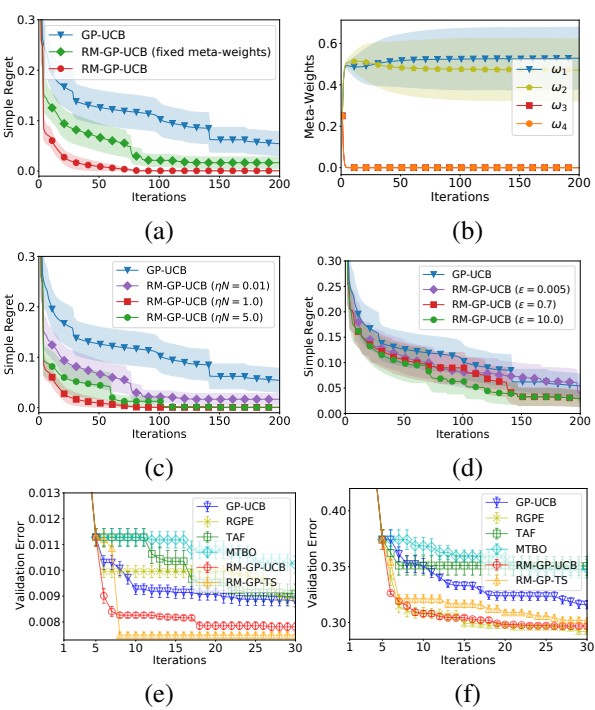

Figure 1: (a) The simple regret and (b) meta-weights optimized by RM-GP-UCB. The impact of (c) $\eta$ and (d) $\epsilon$. Best validation error of CNN for (e) MNIST and (f) CIFAR-10.

## 6.1 SYNTHETIC EXPERIMENTS

We firstly explore the effectiveness of our online meta-weight optimization (Sec. 5) and the impact of different algorithmic parameters by optimizing synthetic functions drawn from GPs. For each objective function, we construct $M = 4$ meta-tasks with $N = N_i = 20$ meta-observations each. The function gaps are chosen as $d_1 = d_2 = 0.05$ and $d_3 = d_4 = 4.0$ such that the last 2 meta-tasks are dissimilar to the target task. Fig. 1a plots the simple regrets averaged over 20 randomly drawn synthetic functions, with $\eta N = 1.0$, $\epsilon = 0.7$, and $r = 0.7$. The figure shows that RM-GP-UCB with online meta-weight optimization significantly outperforms RM-GP-UCB with fixed meta-weights ($\omega_i = 1/4$ for all $i$). Fig. 1b plots the meta-weights optimized by RM-GP-UCB for the red curve in Fig. 1a, showing that the weights given to the last two meta-tasks which are dissimilar to the target task are rapidly reduced. These results verify the effectiveness of online meta-weight optimization in reducing the impact of dissimilar meta-tasks.

We also investigate the impact of $\eta$ and $\epsilon$. Fig. 1c shows the performances of different values of $\eta$, with fixed $\epsilon = 0.7$ and $r = 0.7$. The figure demonstrates that an excessively small $\eta$ (purple curve) negatively impacts the performance, since RM-GP-UCB is unable to quickly reduce the weights of dissimilar meta-tasks (Fig. 4a in Appendix D.1). Moreover, an overly large $\eta$ is also slightly detrimental (green curve) since

it rapidly assigns a large weight to one of the two useful meta-tasks (Fig. 4c in Appendix D.1), thus failing to utilize the other useful meta-task. Fig. 1d illustrates the impact of $\epsilon$ when all function gaps are large: $d_i = 8.0$ for all $i$.[5] The figure shows that even when all meta-tasks are dissimilar, our adaptive selection of $\nu_t$ is able to diminish their negative impact and allow RM-GP-UCB to perform comparably to GP-UCB. Furthermore, in this adverse scenario, a faster decline of the impact of the meta-tasks (i.e., faster decay of $\eta_t$ via larger $\epsilon$) leads to slightly better performance.

## 6.2 REAL-WORLD EXPERIMENTS

**Hyperparameter Tuning for Convolutional Neural Networks (CNNs).** We apply meta-BO to hyperparameter tuning of ML models with the previous tasks using other datasets as the meta-tasks. We tune 3 hyperparameters of CNNs using 4 widely used image datasets: MNIST, SVHN, CIFAR-10 and CIFAR-100. Specifically, in each experiment, one of the four datasets is selected to produce the target function $f$ which maps a hyperparameter setting to a validation accuracy obtained using this dataset. The meta-observations are generated from 3 independent BO tasks (each with 50 iterations) using the other 3 datasets, i.e., $M = 3$ and $N_i = 50$ for $i = 1, 2, 3$ in all 4 experiments. The results for MNIST and CIFAR-10 are plotted in Figs. 1e and 1f while the remaining results are shown in Appendix D.2 (Fig. 6). The results show that RM-GP-UCB is the only method that consistently performs well in all tasks, and that RM-GP-TS performs much better than RM-GP-UCB (and other methods) for MNIST, yet worse in the other tasks. We have also adopted the Omniglot dataset [Lake et al., 2015] commonly used in meta-learning, for which RM-GP-UCB performs the best (Fig. 7, Appendix D.2).

**Non-stationary Bayesian Optimization.** Meta-BO can be naturally applied to non-stationary BO problems in which the unknown objective function evolves over time since the previous (outdated) observations can be treated as the meta-observations. We consider here automated ML for clinical diagnosis. As the data from new patients becomes available regularly, clinicians often need to periodically update the dataset and re-run hyperparameter optimization for the ML model used for clinical diagnosis. This stimulates the question as to whether the previous hyperparameter tuning tasks using the outdated patients data can help accelerate the current task. We consider the problem of diabetes prediction [Smith et al., 1988] with *logistic regression* (LR) and tune 3 LR hyperparameters. We create 5 progressively growing datasets (including the full dataset), treating (the hyperparameter tuning task using) the full dataset as the target task and the 4 smaller datasets as the meta-tasks. Specifically, the entire dataset consists of 768 data instances,

among which 77 instances are set aside to measure the validation accuracy. The sizes of the 5 progressively growing training datasets (i.e., corresponding to the 4 meta-tasks and the target task, respectively) are 138, 276, 414, 552, and 691. The results (Fig. 2a) show that RM-GP-TS outperforms all other methods in this task. Moreover, we also compare the runtime of different methods in Fig. 2b: RM-GP-TS is significantly more efficient than all other methods, and the methods building separate GP surrogates for different tasks (i.e. RM-GP-UCB, RGPE and TAF) are more efficient than MTBO which includes all observations in a single GP (Sec. 1).

**Hyperparameter Tuning for Support Vector Machines (SVMs).** We also tune the hyperparameters of SVMs using a tabular benchmark dataset [Wistuba et al., 2015a] which has also been adopted by RGPE [Feurer et al., 2018]. The benchmark was constructed by evaluating a fixed grid of 288 SVM hyperparameter configurations using 50 *diverse* datasets (i.e., containing many dissimilar tasks). We follow the setting used by RGPE [Feurer et al., 2018]: In every trial, we fix one of the tasks as the target task, and the remaining $M = 49$ tasks as the meta-tasks; for every meta-task $i$, we randomly select $N_i = 50$ hyperparameter configurations as the meta-observations. The results in Fig. 2c show that our RM-GP-UCB performs comparably to RGPE, outperforming the other methods; RM-GP-TS performs unsatisfactorily in this experiment with diverse tasks. Of note, this experiment has the most favorable setting for PEM-BO [Wang et al., 2018] because (a) PEM-BO has been shown to require a massive set of meta-observations ($\geq 5000$) to perform well [Wang et al., 2018], and this experiment has the largest number ($49 \times 50 = 2450$) of meta-observations among all experiments; (b) the domain here is discrete, which is much easier for the application of PEM-BO.

**Human Activity Recognition (HAR).** HAR using mobile devices has promising applications in various domains such as healthcare [Reyes-Ortiz et al., 2013]. When optimizing the configurations (hyperparameters) of the activity prediction model (ML model) for a subject, the previous optimization tasks for other subjects might be helpful. However, cross-subject transfer in HAR is challenging due to high *individual variability* [Soleimani and Nazerfard, 2019], which makes HAR suitable for evaluating the robustness of a meta-BO algorithm against dissimilar meta-tasks. We use the data collected through mobile phones from 30 subjects performing 6 activities and use *support vector machines* (SVM) for activity prediction. Every task corresponds to tuning 2 SVM hyperparameters for a subject. We run a separate BO (30 iterations) for each of the 21 subjects to generate the meta-observations ($M = 21$, $N_i = 30$ for $i = 1, \dots, 21$) and use the other 9 subjects for validation. The results are shown in Fig. 2d (averaged over the 9 subjects, each further averaged over 5 random initializations), in which RM-GP-UCB delivers the best performance, followed by RGPE; RM-GP-TS

---

[5]We use $\eta = 1/N$ and fix $r$ at a large value (0.99) so that the decaying rate of $\nu_t$ is purely decided by $\epsilon$.

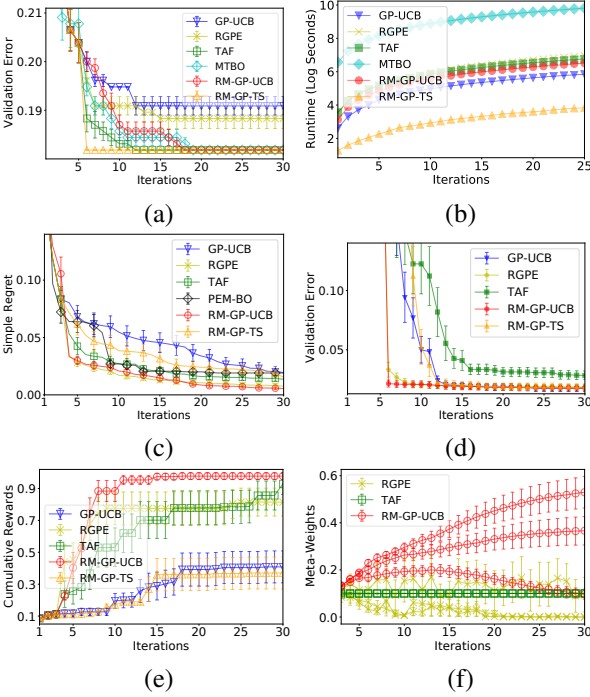

Figure 2: (a) Best validation error of LR for diabetes diagnosis. (b) Runtime in non-stationary BO experiment. (c) Simple regret on SVM benchmark. (d) Best validation errors for HAR. (e) Best cumulative rewards and (f) learned meta-weights for the 3 similar meta-tasks for the RL experiment.

again fails to perform effectively, suggesting that it is less robust against the individual variability in HAR.

**Policy Search for Reinforcement Learning (RL).** When optimizing the RL policy of an agent in an environment, the agent's experience in other related environments may help to make learning more efficient [Duan et al., 2016, Wang et al., 2016]. We apply meta-BO to policy search in RL to maximize the cumulative rewards in an episode, using the Cart-Pole environment from OpenAI Gym [Brockman et al., 2016] with 8 policy parameters. We simulate different environments by setting the agent to different initial states. In particular, we choose $M = 10$ different initial states, among which the majority (i.e., 7) are randomly generated (i.e., dissimilar meta-tasks) and the other 3 are designed to be close to the initial state of the target task so that they are similar to the target task. An independent BO task with 50 iterations is run for every initial state, i.e., $N_i = 50$ for $i = 1, \ldots, 10$. Figs. 2e and 2f plot the (normalized) cumulative rewards of different algorithms and their learned meta-weights for the 3 similar meta-tasks. The results show that RM-GP-UCB achieves the best performance (Fig. 2e), and it is more effective than RGPE and TAF at identifying the 3 similar meta-tasks (Fig. 2f). RGPE and TAF fail to correctly identify similar meta-tasks because they learn the meta-weights based on how accurately each GP surrogate predicts the

*pairwise ranking of the target observations* (more details in Sec. 7). However, in the Cart-Pole environment, many target observations have equal values, which confuses the pairwise ranking and makes the learned meta-weights unreliable. RM-GP-TS again only performs comparably with standard GP-UCB (Fig. 2e).

## 6.3 EXPERIMENTAL DISCUSSION

In most experimental results (Figs. 1 and 2), the performance advantage of RM-GP-UCB is most evident at the initial stage. This is likely to corroborate our theoretical insights that the meta-tasks can help improve the convergence of RM-GP-UCB at the initial stage by reducing the degree of exploration (Sec. 4.1). A potential limitation of our online meta-weight optimization (Sec. 5) is that it does not account for the scenario where the meta-functions are shifted or scaled versions of the target function. However, note that in some scenarios, the scale of the meta-functions is informative about task similarity and thus should not be removed. For example, in our clinical diagnosis (i.e., non-stationary BO) experiment, the more recently completed meta-tasks (with larger training set, smaller validation errors, and thus smaller function gaps) are expected to be more similar to the target task. Furthermore, as demonstrated by the green curve in Fig. 1a, in some cases, even though the meta-weights are not optimized, RM-GP-UCB still performs favorably. This implies its robustness against mis-specification of the meta-weights.

RM-GP-UCB is the only method that consistently outperforms standard GP-UCB in *all* experiments (Figs. 1 and 2), whereas other methods perform either comparably with or worse than GP-UCB in some experiments (e.g., RGPE in Figs. 1e and 2a, TAF in Figs. 1e, 1f and 2d). This might be attributed to RM-GP-UCB's theoretically guaranteed robustness against dissimilar meta-tasks (Sec. 4) and its ability to diminish their impact in a principled way (Sec. 5). In particular, RM-GP-UCB performs significantly better than RM-GP-TS in those experiments with a large number of dissimilar meta-tasks (Figs. 2c-e), which may be explained by RM-GP-UCB's better theoretically guaranteed robustness against dissimilar meta-tasks than RM-GP-TS (Sec. 4.2). However, Figs. 1e-f and Fig. 2a show that RM-GP-TS performs competitively in some experiments with more favorable settings (i.e., less dissimilar meta-tasks), which might result from the repeatedly observed empirical effectiveness of TS-based algorithms [Chapelle and Li, 2011, Russo et al., 2017]. Moreover, the computational efficiency of RM-GP-TS is markedly superior to other methods (Fig. 2b). These theoretical and empirical comparisons between RM-GP-UCB and RM-GP-TS may provide useful insights for other meta-BO algorithms and potentially for a broader range of problems (e.g., meta-learning for multi-armed bandits and RL) in terms of the relative strengths and weaknesses of

UCB- and TS-based algorithms.

# 7 RELATED WORKS

Some previous works on meta-BO build a joint GP surrogate using all previous and current observations, and represent task similarity through meta-features [Bardenet et al., 2013, Schilling et al., 2016, Yogatama and Mann, 2014]. However, these algorithms suffer from the requirement of handcrafted meta-features, which is avoided in other works that learn task similarity from the observations [Swersky et al., 2013, Shilton et al., 2017]. For example, multitask BO [Swersky et al., 2013] uses a multitask GP as a surrogate and models each task as an output of the GP. These works include all previous and current observations in a single GP surrogate and are thus limited by the scalability of GPs. There have also been other empirical works which replace GP by Bayesian linear regression for scalability [Perrone et al., 2018], tackle sequentially arriving tasks [Golovin et al., 2017, Poloczek et al., 2016], learn a set of good initializations [Feurer et al., 2015, Wistuba et al., 2015b], learn a reduced search space for BO from previous tasks [Perrone et al., 2019], handle the issue of different function scales using Gaussian Copulas [Salinas et al., 2020], learn the task similarities through the distance between the distributions of the optima from different tasks [Ramachandran et al., 2018], or use the meta-observations to learn the entire acquisition function through RL [Volpp et al., 2020]. Wang et al. [2018] have learned the GP prior from previous tasks and given theoretical guarantees. However, they have shown in both theory and practice that a large training set of meta-observations ($\geq 5000$) is required for their method to work well, while we focus on the more practical setting of meta-BO where the number of available meta-observations may be small. We have also verified that our algorithm outperforms the method from Wang et al. [2018] in the experiment that is most favorable for their method among all our experiments (more details in the third paragraph of Sec. 6.2). Meta-BO is also related to the works on multi-fidelity BO [Dai et al., 2019, Kandasamy et al., 2016, Poloczek et al., 2017, Wu et al., 2020, Zhang et al., 2020, 2017], since the previous tasks can be viewed as low-fidelity functions which can approximate the target function and are cheap to query. However, multi-fidelity BO allows querying the low-fidelity functions during the BO process, whereas meta-BO algorithms can only query the target function, i.e., the highest-fidelity function. Moreover, meta-BO is also related to the previous works on BO which involve multiple agents (i.e., analogous to multiple tasks in meta-BO), such as federated BO [Dai et al., 2020b, 2021, Sim et al., 2021] or BO methods based on game-theoretical approaches [Dai et al., 2020a, Sessa et al., 2019].

Some works have aimed to improve the scalability of GP-based meta-BO algorithms by building a separate GP surrogate for each task [Feurer et al., 2018, Wistuba et al., 2016, 2018]. Wistuba et al. [2016] use a weighted combination of the posterior mean of each individual GP surrogate as the joint posterior mean while the posterior variance is derived using only the target observations. RGPE [Feurer et al., 2018] has extended the work of Wistuba et al. [2016] by estimating the joint objective function as a weighted combination of individual objective functions, such that the resulting joint surrogate remains a GP (unlike Wistuba et al. [2016]) and can thus be plugged into standard BO algorithms. Note that RGPE differs from our RM-GP-UCB algorithm in that RGPE uses a weighted combination of individual GP surrogates to derive a joint GP surrogate, whereas our RM-GP-UCB leverage a weighted combination of individual acquisition functions. Wistuba et al. [2018] have proposed TAF, which also uses a weighted combination of the acquisition functions (i.e., expected improvement) from the individual tasks for query selection. In these works, the weight of a previous task is heuristically chosen to be proportional to the accuracy of the *pairwise ranking of the target observations* produced by either (a) the posterior mean of the GP surrogate of the previous task (TAF) [Wistuba et al., 2018] or (b) functions sampled from the posterior GP surrogate (RGPE) [Feurer et al., 2018].

# 8 CONCLUSION

We have introduced RM-GP-UCB and RM-GP-TS, both of which are asymptotically no-regret even if all meta-tasks are dissimilar to the target task. We leverage the theoretical results to learn the task similarities in a principled way via online learning. Theoretical and empirical comparisons show that RM-GP-UCB is more robust against dissimilar tasks, whereas RM-GP-TS performs effectively in more favorable cases and is more computationally efficient.

**Acknowledgements**

This research/project is supported by A*STAR under its RIE2020 Advanced Manufacturing and Engineering (AME) Industry Alignment Fund – Pre Positioning (IAF-PP) (Award A19E4a0101) and by the Singapore Ministry of Education Academic Research Fund Tier 1. This research is part of the programme DesCartes and is supported by the National Research Foundation, Prime Minister's Office, Singapore under its Campus for Research Excellence and Technological Enterprise (CREATE) programme.

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
