# OpenReview forum: "On Provably Robust Meta-Bayesian Optimization"
_auai.org/UAI/2022/Conference — UAI 2022 Poster_

### Official Review · Reviewer_13HA · 2022-04-12

**Q2(1) Originality/Novelty:** 3
**Q2(2) Significance/Impact:** 3
**Q2(3) Correctness/Technical Quality:** 3
**Q2(6) Clarity Of Writing:** 4
**Q6 Overall Score:** 7
**Q8 Confidence In Your Score:** 5

**Q1 Summary And Contributions:**

Conventional BO can only handle each task independently. However, when providing a set of tasks and there exist correlations among the tasks, treating them independently are waste of computing resources and result in inferior performance. Powered by the idea of meta-learning, meta-BO is a relatively new topic. The key idea is how to utilize the correlations of tasks to accelerate the performance of new BO tasks. The authors proposed two provable no-regret meta-BO algorithms.

**Q2 Assessment Of The Paper:**

More detailed information regarding each of these aspects is given below:

**Q2(4) Quality Of Experiments (Optional):**

3: Good: The experimental evaluation is adequate, and the results convincingly support the main claims.

**Q2(5) Reproducibility:**

3: Good: Key resources (e.g., proofs, code, data) are available and key details (e.g., proofs, experimental setup) are sufficiently well-described for competent researchers to confidently reproduce the main results.

**Q3 Main Strengths:**

Meta-BO is a relatively new topic. The study given in this paper is novel for meta-BO. First, the authors gave theoretical justifications for the no-regret guarantee of their two proposed meta-BO algorithms. Second, the previous meta-BO methods either brutally use the dataset distance to transfer the task or evaluate the dataset distributions. In this word, the authors also provide a principled way to infer the task weight.

**Q4 Main Weakness:**

The material presented in this paper is self-contained and the study given is comprehensive enough. However, though the main surrogate used for BO is GP, there are lots of existing works that use BNN as the surrogate. I am sure that it is hard to provide theoretical guarantees for BNN based BO. But it will be better to include other meta-BO baselines that might use BNN as the surrogate.

**Q5 Detailed Comments To The Authors:**

I have no further comment or criticism on this work. I think the authors have done good work. The experiment parts are also detailed, not only including synthetic functions, but also real-world hyper-parameter tunning applications.

**Q7 Justification For Your Score:**

Meta-BO is an important topic due to the BO itself is a very expensive procedure. Inferring prior knowledge from similar tasks can accelerate the current task is crucial. Previous work did not address or formulate the meta-BO problem in a principled and provable way. This is the main strength of this paper.

**Q9 Complying With Reviewing Instructions:**

1: Yes.

---

### Official Review · Reviewer_1u3v · 2022-04-12

**Q2(1) Originality/Novelty:** 3
**Q2(2) Significance/Impact:** 3
**Q2(3) Correctness/Technical Quality:** 3
**Q2(6) Clarity Of Writing:** 4
**Q6 Overall Score:** 8
**Q8 Confidence In Your Score:** 4

**Q1 Summary And Contributions:**

Proposes and tests transfer-learning (or multi-task learning) method in Bayesian optimization (BO).  Demonstrates theoretically "no regret" property (of a close approximation of the algorithm used) and compares algorithm favorably to other BO and meta-BO algorithms.

**Q2 Assessment Of The Paper:**

More detailed information regarding each of these aspects is given below:

**Q2(4) Quality Of Experiments (Optional):**

4: Excellent: The experimental evaluation is comprehensive and the results are compelling.

**Q2(5) Reproducibility:**

4: Excellent: Key resources (e.g., proofs, code, data) are available and key details (e.g., proof sketches, experimental setup) are comprehensively described for competent researchers to confidently and easily reproduce the main results.

**Q3 Main Strengths:**

+ Proposed method shown to work well on a variety of tasks and against a fair set of "competing" methods
+ No-regret proof
+ Well presented and written

**Q4 Main Weakness:**

- Bound used to optimize the weight of each "prior task," yet direct optimization of this bound would result in 0 weight to these tasks.  Suggests bound does not capture necessary characteristics for this purpose.
- No-regret is not surprising given form of the acquisition function (essentially, it converges to a single-task acquisition function with known no-regret properties as T -> infinity)

**Q5 Detailed Comments To The Authors:**

My main concern, as mentioned above, is that the bound is used to pick w and v (nu?).  However, truly optimizing the bound wrt to these would lead to no weight being placed on the prior problems.  This bound only captures "differences" of the current problem from the prior problems, not "similarity" --- that is how much they differ rather than what information might be transferable.  So, why should we expect it should be used for setting w/v?  The path taken in this paper has to force the solution not to allow 1) all weight to be placed on a single prior problem and 2) that weight to be set all the way to zero arbitrarily quickly.

Given the assumption that N_i = N (for all i), I would expect an experiment in which this was not true (to test the importance of that assumption).  I didn't see one.  This would be a good addition.

Minor points:
The diabetes problem could use a bit more detail (from the appendix -- say the last sentence plus a bit) on how it was transformed into a meta-learning problem so that the results can be gauged better.

I'm uncertain why you get N^{3/2} instead of N in Equation 7.  I've looked this over multiple times and I feel I must be missing something, but I cannot find it.  I see an approximation of N(sqrt(log(N)) + d) which I might make Nd, but not N^{3/2}d.  Feel free to ignore this comment if I'm just missing something obvious.

Multi-fidelity BO is a closely related concept.  I was surprised one of those acquisition functions was not tried (pinning the "fidelity" to be the current task).   At least it would be useful for the reader new to BO to have a mention of how it differs in one of the related works sections.

Theorem 1 mentions lambda.  I did not see it defined prior to this theorem.  Again, ignore this if I just failed my reading comprehension test.

Given that Equations (2) and (3) use the same notation for the acquisition function (and that the method names look very similar), it was confusing to me at a quick first pass.


**Q7 Justification For Your Score:**

A solid paper on meta-BO.

**Q9 Complying With Reviewing Instructions:**

1: Yes.

---

### Official Review · Reviewer_82m4 · 2022-04-14

**Q2(1) Originality/Novelty:** 3
**Q2(2) Significance/Impact:** 3
**Q2(3) Correctness/Technical Quality:** 3
**Q2(6) Clarity Of Writing:** 3
**Q6 Overall Score:** 7
**Q8 Confidence In Your Score:** 3

**Q1 Summary And Contributions:**

This paper proposed two novel Bayesian Optimization (BO) problems that are capable of leverage previous evaluations of potentially related functions and thus lead to potentially better optimization results. For target function f with bounded norm induced by the kernel of the GP prior, high probability finite time regret upper-bounds are established. The proposed algorithms outperform a number of existing BO algorithms in extensive real-world experiments.

**Q2 Assessment Of The Paper:**

More detailed information regarding each of these aspects is given below:

**Q2(4) Quality Of Experiments (Optional):**

4: Excellent: The experimental evaluation is comprehensive and the results are compelling.

**Q2(5) Reproducibility:**

3: Good: Key resources (e.g., proofs, code, data) are available and key details (e.g., proofs, experimental setup) are sufficiently well-described for competent researchers to confidently reproduce the main results.

**Q3 Main Strengths:**

This paper gives novel meta-BO algorithms with provable robust performance guarantee against harmful dissimilar tasks, which is a technically significant result compared to previous works. The experiment results of the new algorithms look satisfactory on a large variety of real-world hyper-parameter tuning tasks, which suggests they are competitive in practice.


**Q4 Main Weakness:**

Overall this paper is satisfactory to me. Some minor questions are listed in Q5.

**Q5 Detailed Comments To The Authors:**

I have some concerns on whether to emphasize the scalability of the new algorithms. In Theorem 1, N_i’s and d_i’s are assumed to be constants independent with T, then the O(MN^2 + T^2) inference cost is asymptotically not far from O((MN + T)^2) in the single GP case. If we allow N_i’’s to be parameterized in T, then we need additional assumptions on d_i’s, and must take the order of N_i and d_i into consideration when picking nu_t.

**Q7 Justification For Your Score:**

I have read the main paper and skimmed the proof in the appendix. The score is mainly based on the main paper while the proof also looks okay to me.

**Q9 Complying With Reviewing Instructions:**

1: Yes.

---

### Official Review · Reviewer_vGXz · 2022-04-16

**Q2(1) Originality/Novelty:** 3
**Q2(2) Significance/Impact:** 3
**Q2(3) Correctness/Technical Quality:** 3
**Q2(6) Clarity Of Writing:** 3
**Q6 Overall Score:** 6
**Q8 Confidence In Your Score:** 3

**Q1 Summary And Contributions:**

The authors propose a method for meta-learning Bayesian Optimization (meta-BO), that exploits data available from similar tasks while ensuring robustness against potentially harmful dissimilar tasks. Two approaches are proposed: robust meta-Gaussian process-upper confidence bound (RM-GP-UCB) and RM-GP-Thompson sampling (RM-GP-TS). The theory for both algorithms shows asymptotically no regret. RM-GP-UCB enjoys better theoretical robustness than RM-GP-TS.



**Q2 Assessment Of The Paper:**

More detailed information regarding each of these aspects is given below:

**Q2(4) Quality Of Experiments (Optional):**

3: Good: The experimental evaluation is adequate, and the results convincingly support the main claims.

**Q2(5) Reproducibility:**

3: Good: Key resources (e.g., proofs, code, data) are available and key details (e.g., proofs, experimental setup) are sufficiently well-described for competent researchers to confidently reproduce the main results.

**Q3 Main Strengths:**

- Clearly and well-written paper.
- Significant theoretical results for both  RM-GP-UCB and RM-GP-TS.
- The authors can optimize weights assigned to previous tasks to reduce the impact of dissimilar tasks.
- Exhaustive experimental validation.

**Q4 Main Weakness:**

- Proposed method does not account for the scenario where the meta-functions are shifted or scaled versions of the target function.

**Q5 Detailed Comments To The Authors:**

- The plots in the paper are too small to be read clearly. They should be made bigger before the paper can be accepted.

**Q7 Justification For Your Score:**

This is a significant contribution as shown by the exhaustive experiments. The paper is clearly  written and the theoretical
results justify the proposed method.

**Q9 Complying With Reviewing Instructions:**

1: Yes.

---

### Decision · Program_Chairs · 2022-05-15

**Decision:**

Accept (Poster)

**Comment:**

Meta Review: The paper develops a very interesting idea of meta Bayesian optimisation where optimisation over one problem helps to solve another. Reviewers were very positive about this work but also made a number of suggestions. Please take these into account for the final version.